# Three-Month Outcomes in Cancer Patients with Superficial or Deep Vein Thrombosis in the Lower Limbs: Results from the RIETE Registry

**DOI:** 10.3390/cancers15072034

**Published:** 2023-03-29

**Authors:** Philippe Debourdeau, Laurent Bertoletti, Carme Font, Juan José López-Núñez, Covadonga Gómez-Cuervo, Isabelle Mahe, Remedios Otero-Candelera, Maria Dolores Adarraga, Patricia López-Miguel, Manuel Monreal

**Affiliations:** 1Centre Hospitalier Joseph Imbert, BP 80195, 13637 Arles, France; 2Département of Médecine Vasculaire et Thérapeutique, CIC 1408, SAINBIOSE U1059, INSERM, CHU Saint-Étienne, Mines Saint-Etienne, Université Jean Monnet Saint-Étienne, 42000 Saint-Etienne, France; 3Department of Medical Oncology, Hospital Clínic, 08036 Barcelona, Spain; 4Department of Internal Medicine, Hospital Germans Trias i Pujol, 08916 Barcelona, Spainmmonrealb@ucam.com (M.M.); 5Department of Medicine, Universitat Autònoma de Barcelona, 08193 Barcelona, Spain; 6Department of Internal Medicine, Hospital Universitario 12 de Octubre, 28041 Madrid, Spain; 7Department of Internal Medicine, University Paris 7, Hôpital Louis Mourier, 92700 Colombes, France; 8Department of Pneumonology, Hospital Universitario Virgen del Rocío, 41013 Seville, Spain; 9Department of Internal Medicine, Hospital Universitario Reina Sofía, 14004 Córdoba, Spain; 10Department of Pneumonology, Hospital General Universitario de Albacete, 02008 Albacete, Spain; 11Chair for the Study of Thromboembolic Disease, Faculty of Health Sciences, UCAM—Universidad Católica San Antonio de Murcia, Universidad Autónoma de Barcelona, 08193 Barcelona, Spain

**Keywords:** superficial vein thrombosis, deep venous thrombosis, cancers, anticoagulants, prognosis

## Abstract

**Simple Summary:**

The natural history of isolated lower-limb superficial vein thrombosis (SVT) in cancer patients is not well understood. To address this gap, we analyzed data from the RIETE registry, which included 110 cancer patients with isolated SVT, 1695 cancer patients with isolated deep vein thrombosis (DVT), and 1030 non-cancer patients with SVT. Compared to cancer patients with DVT, cancer patients with lower-limb SVT received lower doses of anticoagulants, had similar rates of subsequent pulmonary embolism (PE), DVT or SVT, were less likely to have metastases and had a lower mortality rate. However, cancer patients with lower-limb SVT had a higher rate of subsequent PE, DVT or SVT than non-cancer patients with SVT. Additionally, in cancer patients with lower-limb SVT, almost all recurrences and bleeding occurred within the first three months.

**Abstract:**

Background: The clinical characteristics and outcomes of cancer patients with lower-limb isolated superficial vein thrombosis (SVT) have not been consistently evaluated. Methods: We used data in the RIETE registry to compare the clinical characteristics and 90-day outcomes for patients with: (1) active cancer and lower-limb SVT; (2) active cancer and lower-limb deep vein thrombosis (DVT); (3) lower-limb SVT without cancer. The primary outcomes included subsequent symptomatic SVT, DVT or pulmonary embolism (PE). Secondary outcomes were major bleeding and death. Results: From March 2015 to April 2021, there were 110 patients with cancer and SVT, 1695 with cancer and DVT, and 1030 with SVT but no cancer. Most patients in all subgroups (93%, 99% and 96%, respectively) received anticoagulants, while those with SVT received lower daily doses of low-molecular-weight heparin (114 ± 58, 163 ± 44, and 106 ± 50 IU/kg, respectively). During the first 90 days, 101 patients (3.6%) developed subsequent VTE (PE 47, DVT 41, SVT 13), whereas 72 (2.5%) had major bleeding and 282 (9.9%) died. Among the three groups, 90-day events were, respectively: VTE at rates of 7.3%, 4.0% and 2.4%; major bleeding at rates of 2.7%, 3.9% and 0.3%; mortality at rates of 8.2%, 16% and 0.3%. Between D90 and D180, only one SVT recurrence and one death occurred in SVT cancer patients. In multivariable analysis, cancer was associated with subsequent VTE (HR = 2.04; 1.15–3.62), while initial presentation as SVT or DVT were not associated with a different risk. Conclusions: The risk for subsequent VTE (including symptomatic SVT, DVT or PE) was similar in cancer patients with isolated SVT than in those with isolated DVT.

## 1. Introduction

The association between cancer and venous thromboembolism (VTE) has been known since 1865, when Armand Trousseau reported his own case, and has been widely documented [1]. However, the original report did not refer to a deep venous thrombosis (DVT) but to a superficial vein thrombosis (SVT) that appeared as the first manifestation of gastric cancer [1]. Isolated SVT is defined by the presence of a blood clot in a superficial vein, without concomitant deep vein thrombosis (DVT) or pulmonary embolism (PE) [2]. The natural history of cancer patients with isolated SVT has not been consistently evaluated as for many years it was considered a benign disease despite being thought to occur at least as often as DVT [2,3]. Recent reports suggest that cancer patients presenting with isolated SVT are at increased risk of developing DVT or PE [4], and that the risk for subsequent DVT or PE may be similar in cancer patients with SVT or DVT [5,6].

Current knowledge on the natural history and optimal therapy of cancer patients with isolated SVT is scarce since most studies on this topic have included few patients with cancer. Therefore, in the absence of evidence-based recommendations, many cancer patients with isolated SVT are prescribed sub-therapeutic doses of anticoagulants [6,7]. This is why we decided to compare the clinical characteristics, therapeutic strategies and 3-month outcomes of 3 subgroups of patients included in the Registro Informatizado de la Enfermedad TromboEmbolica (RIETE): (1) active cancer patients with isolated SVT in the lower limbs; (2) active cancer patients with isolated DVT in the lower limbs; and (3) non-cancer patients with isolated lower-limb SVT.

## 2. Methods

### 2.1. Data Source

We used the data from the RIETE registry, which prospectively collects information on patients with symptomatic, objectively confirmed acute VTE (ClinicalTrials.gov identifier, NCT02832245). Previous publications have described the design and the means of conducting collection for the RIETE registry [8]. All patients (or their relatives) provided written or oral informed consent for participation in the registry in accordance with the local ethics committee requirements.

### 2.2. Inclusion Criteria

At each participating site, the RIETE investigators aimed to enroll consecutive patients with acute VTE, cases of which had to be confirmed by objective testing. SVT as a variable has only been included in RIETE since March 2015. SVT was defined as the presence of a confirmed non-compressible venous segment by compression ultrasonography in a superficial vein without concomitant DVT or PE [8]. Patients presenting with both SVT and DVT were considered as having DVT. Patients with upper limb SVT or with symptomatic PE were excluded from the study. Patients participating in a randomized trial with a blind medication were excluded. SVT cancer patients were managed according to each participating hospital’s clinical practice, because there were no treatment recommendations. All participants were followed up for at least 3 months in the outpatient clinic and beyond the third month in cases of participants’ approval

The following parameters were recorded: age, sex, weight, VTE risk factors (recent surgery or immobility ≥4 days, estrogen use, pregnancy or postpartum, prior VTE or SVT, leg varicosities), comorbidities (chronic heart failure, lung disease or renal insufficiency), blood test abnormalities (anemia, leukocytosis, thrombocytopenia), concomitant use of corticosteroids or antiplatelet agents, cancer characteristics (time elapsed from diagnosis of cancer to VTE, site of tumor, metastases and anti-tumor therapy), and treatment (drugs, doses, duration).

### 2.3. Study Design

We compared the clinical characteristics, treatment strategies and 3-month outcomes in patients with: (1) active cancer and isolated lower-limb SVT; (2) active cancer and isolated lower-limb DVT; and (3) isolated lower-limb SVT and no cancer. Only symptomatic and objectively confirmed SVT or DVT events were considered. Active cancer was defined as newly diagnosed cancer (<3 months) or when a patient is receiving anti-neoplastic treatment of any type [9]. The primary outcome was the development of symptomatic, objectively confirmed subsequent VTE events (SVT, DVT or PE) during the first 3 months. Secondary outcomes were major bleeding and all-cause mortality. VTE events were adjudicated by the treating physicians and defined as symptomatic SVT, DVT or PE, and categorization was confirmed on compression ultrasonography, computed tomography angiogram, ventilation/perfusion lung scan or magnetic resonance pulmonary angiography. DVT was categorized as proximal when its upper limit was at or above the popliteal vein and SVT was categorized according to its length (<5 cm, 5–10 cm, >10 cm) and to the distance from the saphenofemoral junction. Major bleeding was defined as retroperitoneal, spinal or intracranial bleeding or any overt bleeding requiring the transfusion of two units or more of blood, whereas fatal bleeding was defined as any death occurring within 10 days of a major bleeding in the absence of an alternative cause of death [8]. Although there is not an independent adjudication committee for RIETE, data quality is regularly monitored for accuracy [8].

### 2.4. Statistical Analysis

Categorical variables were reported as frequency counts (percentages) and compared using the chi-square test (two-sided) and Fisher’s exact test (two-sided). Continuous variables were reported as mean and standard deviation (or median with interquartile range, if not normally distributed), and compared using Student’s *t* test. Odds ratios (ORs) and the corresponding 95% confidence intervals (CI) were calculated.

We compared the risk for subsequent VTE events during the first 90 days, according to the presence or absence of cancer and initial presentation as SVT or DVT. We used logistic regression models to adjust the risk to a number of potential confounders. We included variables which were previously reported to be clinically relevant, as well as covariables identified on bivariate analyses as predictors of subsequent VTE. We included competing risk analysis to evaluate the risk for subsequent VTE, with death being the competing risk. Statistical analyses were conducted with the use of IBM SPSS Statistics (version 25).

## 3. Results

From March 2015 to May 2021, there were 110 patients with cancer and SVT, 1695 with cancer and DVT and 1030 with SVT and no cancer (Table 1).

### 3.1. Cancer Patients with SVT vs. Cancer Patients with DVT

Compared to cancer patients with DVT, those with SVT were less likely to be men and were less likely to have experienced recent immobility or recent surgery, but more likely to have undergone prior SVT or lived with leg varicosities (Table 1). As for the cancer characteristics, patients with SVT were less likely to have metastases at baseline and more likely to have breast or hematologic cancers, and were also less likely to have had bladder cancer (Table 2). Most patients in both subgroups received anticoagulant therapy (93% vs. 99%), mostly with low-molecular-weight heparin (LMWH), for both initial- and long-term therapy. Interestingly, patients with SVT received lower doses of LMWH (for initial- and for long-term therapy) than those initially presenting with DVT (Table 3). Few patients in both subgroups received direct oral anticoagulants or Fondaparinux.

During the first 90 days, 76 patients (4.2%) developed subsequent VTE events (PE 41, DVT 33, SVT 2), 69 (3.8%) suffered major bleeding and 279 (15.5%) died (Table 4). Of these, 4 patients died of PE and 13 died of bleeding. There were no differences in the rates of subsequent VTE events (OR: 1.82; 95%CI: 0.75–3.91) or major bleeding (OR: 0.69; 95%CI: 0.17–2.00), but the mortality rate was half in cancer patients with SVT than that in those with DVT (OR: 0.47; 95%CI: 0.22–0.91).

### 3.2. Cancer Patients with SVT vs. Non-Cancer Patients with SVT

Compared to non-cancer patients with SVT, those with cancer were older and less likely to have prior SVT or leg varicosities, but were more likely to have chronic heart failure, anemia, renal insufficiency or to receive corticosteroids at baseline (Table 1). Most patients in both subgroups (93% vs. 99%) received anticoagulant therapy, and most of those with LMWH did so for both initial and for long-term therapy (Table 3). Patients without cancer received similar daily doses of LMWH than those with cancer, but the proportion of patients receiving Fondaparinux was much higher in patients without cancer than in those with cancer (20% vs. 4.5%).

During the first 90 days, 33 patients (1.8%) developed subsequent VTE events (PE 9, DVT 12, SVT 12), 6 (0.5%) suffered major bleeding and 12 (1.0%) died (Table 4). Of these, 3 patients died of PE and none died of bleeding. Patients with cancer had a much higher rate of subsequent VTE events (OR: 4.92; 95%CI: 1.82–12.4), major bleeding (OR: 9.56; 95%CI: 1.63–56.2) or death (OR: 30.3; 95%CI: 8.41–141) than those without cancer.

In patients with SVT, almost all events occurred within the first three months, except for one SVT recurrence and one death in cancer patients and 3 SVT recurrences and 1 DVT recurrence in non-cancer patients.

### 3.3. VTE Recurrence among the Whole Population

After undergoing multivariable analysis, patients aged < 65 years (hazard ratio [HR]: 2.04; 95%CI: 1.27–3.33), with active cancer (HR: 2.04; 95%CI: 1.15–3.62), leukocytosis (HR: 1.75; 95%CI: 1.10–2.77) or creatinine clearance levels ≥ 60 mL/min, were found to be at increased risk for subsequent VTE events (Table 5). Initial presentation as DVT or SVT was not associated with a different risk for subsequent VTE events.

## 4. Discussion

Our data reveal that the risk for subsequent VTE events (including symptomatic SVT, DVT or PE) during the first 3 months of anticoagulants in cancer patients with isolated SVT in the lower limbs is similar to the risk in cancer patients with DVT. This is important since many clinicians do not consider SVT as a potentially fatal disease in patients with cancer [3]. This may also explain why one in every three patients with SVT in our cohort was prescribed sub-therapeutic doses of anticoagulants (particularly LMWH) for initial and for long-term therapy. Interestingly, three of the six patients who died of PE in our cohort had initially presented with isolated SVT. Thus, our findings suggest that isolated lower-limb SVT should not be underestimated, particularly in patients with cancer. Of note, is the fact that the rate of SVT recurrence, DVT, bleeding and death is the same at three and six months in cancer and non-cancer patients with SVT. The lower rate of subsequent VTE in non-cancer patients with isolated SVT than in those with cancer can likely be explained by the influence of cancer on the risk for VTE, as shown in the multivariable analysis which saw an increased risk in patients with cancer. We also found a lower mortality rate in cancer patients with SVT than in cancer patients with DVT. This can likely be explained because those with SVT were less likely to have metastatic cancer, as was also reported in the OPTIMEV study [6].

Most randomized trials of SVT therapy did not include patients with cancer [10,11,12,13,14,15,16], or only included several [17,18,19]. The lack of specific data leads to inconsistency in the current recommendations for the treatment of patients with isolated SVT: different doses (from prophylactic to therapeutic doses) and different durations (from 6 weeks to 6 months) [20,21]. Since we found that almost all thrombotic and all bleeding events occurred within the first three months in cancer patients with isolated SVT, and that those aged <65 years with leukocytosis or normal renal function were at increased risk for subsequent VTE events, our findings may help researchers to design randomized studies which compare different doses and durations of anticoagulants with the aim of reducing the risk for these complications.

In the literature, the risk of the extension of an isolated SVT into the deep venous system is not negligible in patients with cancer. Cancer was associated with a higher risk of developing subsequent DVT or PE in a retrospective study of 411 patients with isolated SVT [4]. In the French retrospective series, 13 of 94 (14%) cancer patients with isolated SVT developed subsequent DVT [7]. In the OPTIMEV series, the rate of subsequent DVT or PE events was also similar for cancer patients with isolated SVT or DVT [6]. It is important to highlight that the risk of developing subsequent DVT or PE in our cohort was independent of receiving lower-than-recommended doses of anticoagulants.

Unexpectedly, only 4.5% of cancer patients with SVT and 20% of non-cancer patients with SVT were prescribed Fondaparinux in our cohort. Fondaparinux is the only drug approved by the European Medicines Agency for the treatment of patients with isolated SVT. Unfortunately, our cohort has not been designed to compare the effectiveness and safety of the different anticoagulants used for SVT therapy.

Finally, we found isolated SVT to be less frequently diagnosed than DVT in patients with cancer. The incidence of SVT in cancer patients has been found to be the same as that of DVT in a literature review [3], but lower in other studies [4,5,12,22]. Since the POST study revealed that one in every 4 cancer patients with SVT may also have concomitant DVT [23], the low frequency of cancer patients with isolated SVT in our cohort may have been due to the higher risk for it with concomitant DVT.

Our study has a number of limitations. First, RIETE is an observational registry and cannot be used to compare therapeutic strategies; RIETE generates hypotheses and may only assist in identifying gaps where well-designed randomized trials could be performed. Second, the length of SVT and its proximity to the sapheno-femoral junction was not available in some patients. Third, the lower proportion of patients with SVT than DVT in cancer patients may reflect under-reporting of SVT due to an underestimation of the severity of this disease. Fourth, our patient sample may not be representative of the cancer population as most oncologists are not involved in the RIETE registry. Nevertheless, our cohort is the largest cohort of patients on this topic and contributes brings data that could help to improve the treatment of isolated SVT in cancer patients. Finally, the treatment of our patients was quite variable, with methods ranging from LMWH to VKAs or DOACs, and the different treatments could have influenced the outcomes in addition to the differences between SVT or DVT.

## 5. Conclusions

In conclusion, the natural history of SVT in cancer patients seems to be better than that of DVT in cancer patients, but worse than that of SVT in non-cancer patients (Figure 1). Our study reveals that the risk of subsequent VTE events in cancer patients with SVT is not lower than in cancer patients with DVT. The risk in non-cancer patients with isolated SVT seems to be lower. However, given the results of the multivariate analysis, this may be due to the influence of their clinical characteristics.

## Figures and Tables

**Figure 1 cancers-15-02034-f001:**
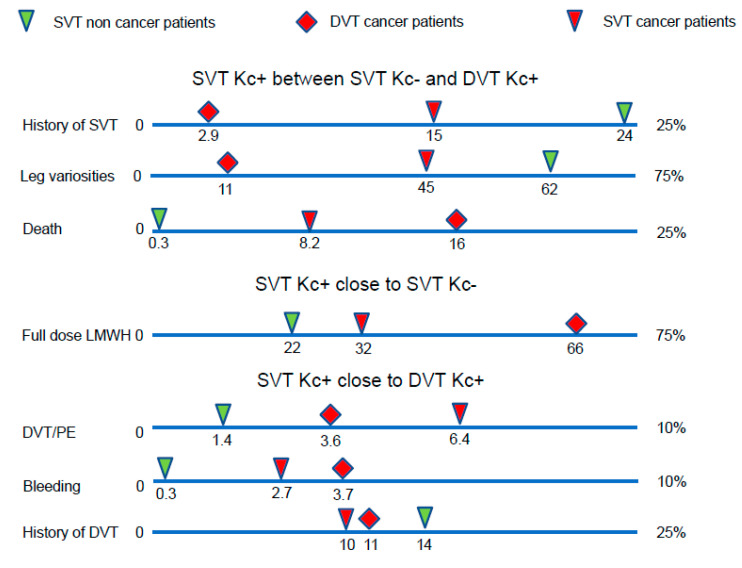
The outcome of SVT in cancer patients was worse than SVT in non-cancer patients, but better than DVT in cancer patients. DVT: deep vein thrombosis, Kc: cancer, LMWH: low molecular weight heparins, PE: pulmonary embolism.

**Table 1 cancers-15-02034-t001:** Clinical characteristics.

	Cancer and Lower-Limb SVT	Cancer and Lower-Limb DVT	Lower-Limb SVT, No Cancer
** *Patients, N* **	** *110* **	** *1695* **	** *1030* **
**Clinical characteristics,**			
Male sex	45 (41%)	860 (51%) *	412 (40%)
Age (mean years ± SD)	67 ± 13	68 ± 13	59 ± 16 ^‡^
Body weight (mean kg ± SD)	76 ± 18	73 ± 15 *	81 ± 17 ^†^
Inpatients	10 (9.3%)	431 (26%) ^‡^	51 (5.1%)
**Initial presentation,**			
*DVT characteristics,*			
Proximal involvement	65 (59%)	1405 (83%) ^‡^	618 (60%)
*SVT characteristics,*			
Length of SVT < 5 cm	19 (24%)	-	179 (22%)
Length of SVT 5–10 cm	30 (38%)	-	311 (39%)
Length of SVT > 10 cm	29 (37%)	-	311 (39%)
Distance from SF junction < 3 cm	34 (36%)	-	291 (33%)
Distance from SF junction > 3 cm	60 (64%)	-	578 (67%)
SVT on a varicose vein	45 (43%)	-	587 (60%) ^†^
**Additional risk factors for VTE,**			
Recent surgery	5 (4.5%)	226 (13%) ^†^	66 (6.4%)
Recent immobility ≥ 4 days	11 (10%)	302 (18%) *	59 (5.7%)
Estrogen use	11 (10%)	133 (7.8%)	49 (4.8%) *
Pregnancy or postpartum	0	2 (0.1%)	40 (3.9%) *
None of the above	83 (75%)	1074 (63%) *	820 (80%)
Prior VTE	11 (10%)	192 (11%)	142 (14%)
Prior SVT	16 (15%)	50 (2.9%) ^‡^	244 (24%) *
Leg varicosities	49 (45%)	185 (11%) ^‡^	643 (62%) ^‡^
**Underlying conditions,**			
Chronic heart failure	5 (4.5%)	55 (3.2%)	14 (1.4%) *
Chronic lung disease	8 (7.3%)	133 (7.8%)	59 (5.7%)
Recent major bleeding	1 (0.9%)	43 (2.5%)	4 (0.4%)
Anemia	46 (42%)	1072 (63%) ^‡^	137 (13%) ^‡^
Leukocyte count > 11,000/µL	17 (17%)	368 (22%)	87 (9.3%) *
Platelet count < 100,000/µL	3 (2.9%)	101 (6.0%)	11 (1.2%)
CrCl levels < 60 mL/min	30 (27%)	523 (31%)	114 (11%) ^‡^
**Concomitant drugs,**			
Corticosteroids	15 (14%)	281 (17%)	44 (4.3%) ^‡^
Antiplatelets	14 (13%)	213 (13%)	111 (11%)

Comparisons between cancer patients with SVT vs. other subgroups: * *p* < 0.05; ^†^
*p* < 0.01; ^‡^
*p* < 0.001. *Abbreviations:* CrCl, creatinine clearance; DVT, deep vein thrombosis; SD, standard deviation; SF, saphenofemoral; SVT, superficial vein thrombosis; VTE, venous thromboembolism.

**Table 2 cancers-15-02034-t002:** Cancer characteristics.

	Cancer and Lower-Limb SVT	Cancer and Lower-Limb DVT
** *Patients, N* **	** *110* **	** *1695* **
**Time from cancer diagnosis**		
Mean months (±SD)	32 ± 62	23 ± 44
Median months (IQR)	6 (0–280)	4 (0–26)
**Metastases**		
Yes	48 (44%)	944 (56%) *
**Sites of cancer,**		
Breast	26 (24%)	202 (12%) ^‡^
Colorectal	16 (15%)	229 (14%)
Lung	11 (10%)	224 (13%)
Hematologic	18 (16%)	141 (8.3%) ^†^
Prostate	6 (5.5%)	127 (7.5%)
Uterine	5 (4.5%)	100 (5.9%)
Stomach	6 (5.5%)	60 (3.5%)
Pancreas	4 (3.6%)	107 (6.3%)
Ovary	3 (2.7%)	85 (5.0%)
Carcinoma of unknown origin	3 (2.7%)	21 (1.2%)
Bladder	2 (1.8%)	117 (6.9%) *
Cerebral	1 (0.9%)	65 (3.8%)
Biliary tract	2 (1.8%)	16 (0.9%)
Melanoma	2 (1.8%)	22 (1.3%)
Other	2 (1.8%)	140 (8.6%)
**Therapy for cancer,**		
Chemotherapy	53 (50%)	805 (52%)
Radiotherapy	15 (14%)	229 (15%)
Immunotherapy	4 (5.3%)	58 (6.7%)
Hormonal therapy	20 (19%)	193 (13%)
None of the above	34 (31%)	635 (37%)

Comparisons between subgroups: * *p* < 0.05; ^†^
*p* < 0.01; ^‡^
*p* < 0.001. *Abbreviations:* SVT, superficial vein thrombosis; DVT, deep vein thrombosis; SD, standard deviation; IQR, inter-quartile range.

**Table 3 cancers-15-02034-t003:** Treatment strategies.

	Cancer and Lower-Limb SVT	Cancer and Lower-Limb DVT	Lower-Limb SVT, No Cancer
** *Patients, N* **	** *110* **	** *1695* **	** *1030* **
**Duration of anticoagulant therapy,**			
Mean days (±SD)	125 ± 155	201 ± 213 ^‡^	125 ± 171
Median days (IQR)	95 (48–145)	128 (86–240)	89 (47–124)
**Initial therapy, N**			
Low-molecular-weight heparin	97 (88%)	1532 (90%)	721 (70%) ^‡^
Mean daily doses (IU/kg/day)	114 ± 58	163 ± 44 ^‡^	106 ± 50 559 (78%) *
LMWH doses < 150 IU/kg/day	66 (68%)	519 (34%) ^‡^	
Fondaparinux	5 (4.5%)	33 (1.9%)	203 (20%) ^‡^
Daily doses < 7.5 mg/daily	4 (3.6%)	1 (0.1%) ^‡^	144 (14%) ^‡^
Rivaroxaban	3 (2.7%)	28 (1.7%)	41 (4.0%)
Apixaban	1 (0.9%)	9 (0.5%)	22 (2.2%)
DOACs at lower-than recommended doses	2 (1.8%)	4 (0.2%) *	12 (1.2%)
Thrombolytic drugs	0	0	1 (0.10%)
Inferior vena cava filter	5 (4.5%)	115 (6.8%)	0
Mechanical thrombolysis	0	9 (0.6%)	3 (0.4%)
No initial therapy	0	12 (0.7%)	15 (1.5%)
**Long-term therapy, N**			
Low-molecular-weight heparin	80 (73%)	1213 (72%)	441 (43%) ^‡^
Mean daily doses (IU/kg/day)	118 ± 60	152 ± 41 ^‡^	97 ± 41 ^†^ 384 (87%) ^‡^
LMWH doses < 150 IU/kg/day	52 (65%)	549 (45%) ^‡^	
Fondaparinux	6 (5.5%)	27 (1.6%) *	175 (17%) ^‡^
Daily doses < 7.5 mg/daily	3 (2.7%)	1 (0.1%) ^‡^	132 (13%) ^‡^
Vitamin K antagonists	6 (5.5%)	177 (10%)	138 (13%) *
DOACs	5 (5%)	159 (9.4%)	145 (14%)
DOACs at lower-than recommended doses	2 (1.8%)	15 (0.9%) ^‡^	14 (1.4%)
No long-term therapy	8 (7.3%)	20 (1.2%) ^‡^	95 (9.2%)

Comparisons between cancer patients with SVT vs. other subgroups: * *p* < 0.05; ^†^
*p* < 0.01; ^‡^
*p* < 0.001. *Abbreviations:* DOACs, direct oral anticoagulants; DVT, deep vein thrombosis; IQR, inter-quartile range; IU, international units; SD, standard deviation; SVT, superficial vein thrombosis; LMWH, low-molecular-weight heparins.

**Table 4 cancers-15-02034-t004:** Outcomes.

	SVT with Cancer		DVT with Cancer		SVT without Cancer	
	*Day 1–90* *N* *110*	*Day 1–180* *N 110*	*Day 1–90* *N 1695*	*Day 1–180* *N 1695*	*Day 1–90* *N 1030*	*Day 1–180* *N 1030*
SVT recurrences	1 (0.9%)	2 (1.8%)	1 (0.1%)	2 (0.1%) *	11 (1.1%)	14 (1.4%)
DVT recurrences	4 (3.6%)	4 (3.6%)	29 (1.7%)	43 (2.5%)	8 (0.8%) *	9 (0.9%) *
PE recurrences	3 (2.7%)	3 (2.7%)	38 (2.2%)	43 (2.5%)	6 (0.6%) *	6 (0.6%) *
VTE recurrences	7 (6.4%)	7 (6.4%)	61 (3.6%)	76 (4.5%)	14 (1.4%)^†^	15 (1.5%)^†^
Major bleeding	3 (2.7%)	3 (2.7%)	66 (3.9%)	81 (4.8%)	3 (0.3%) *	3 (0.3%) *
Overall death	9 (8.2%)	10 (9.1%)	270 (16%) *	359 (21%) ^†^	3 (0.3%) ^‡^	3 (0.3%) ^‡^
Fatal PE	1 (0.9%)	1 (0.9%)	3 (0.2%)	3 (0.2%)	2 (0.2%)	2 (0.2%)
Fatal bleeding	0	0	13 (0.8%)	14 (0.8%)	0	0

Comparisons between cancer patients with SVT vs. other subgroups: * *p* < 0.05; ^†^
*p* < 0.01; ^‡^
*p* < 0.001. *Abbreviations:* DVT, deep vein thrombosis; PE, pulmonary embolism; SVT, superficial vein thrombosis; VTE, venous thromboembolism.

**Table 5 cancers-15-02034-t005:** Uni- and multivariable analysis for VTE recurrences.

	Univariable Analysis	Logistic Regression	Competing Risk Analysis
**Clinical characteristics,**			
Age < 65 years	2.27 (1.45–3.45) ^†^	2.08 (1.28–3.33) ^†^	2.04 (1.27–3.33) ^†^
Male sex	1.35 (0.90–2.03)	-	-
**Proximal DVT**	0.79 (0.51–1.21)	-	-
**Risk factors,**			
Unprovoked VTE	Ref.	Ref.	Ref.
Cancer	1.80 (1.06–3.04) *	2.30 (1.29–4.10) ^†^	2.04 (1.15–3.62) *
Transient risk factors	1.54 (0.63–3.73)	-	-
**Prior VTE,**			
Prior DVT or PE	1.05 (0.56–1.94)	-	-
Prior SVT	1.07 (0.56–2.03)	-	-
Leg varicosities	0.66 (0.41–1.08)	-	-
**Underlying conditions,**			
Chronic heart failure	1.68 (0.60–4.69)	-	-
Chronic lung disease	1.58 (0.81–3.09)	-	-
Recent major bleeding	0.61 (0.08–4.47)	-	-
Anemia	0.87 (0.58–1.32)	-	-
WBC > 11,000/µL	1.93 (1.21–3.07) ^†^	1.73 (1.07–2.78) *	1.75 (1.10–2.77) *
PlC < 100,000/µL	1.04 (0.38–2.89)	-	-
CrCl levels ≥ 60 mL/min	2.70 (1.39–5.26) ^†^	2.33 (1.16–4.76) *	2.27 (1.10–4.55) *
**Concomitant drugs,**			
Corticosteroids	0.96 (0.51–1.82)	-	-
Antiplatelets	1.07 (0.58–1.98)	-	-
**Full-dose anticoagulation**	1.14 (0.75–1.72)	-	-

* *p* < 0.05; ^†^
*p* < 0.01. *Abbreviations:* VTE, venous thromboembolism, DVT, deep vein thrombosis; PE, pulmonary embolism; SVT, superficial vein thrombosis; WBC, white blood cell count; PlC, platelet count; CrCl, creatinine clearance; Ref., reference.

## Data Availability

The data presented in this study are available on request from RIETE registry. The data are not publicly available because the RIETE registry is not a public access registry.

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
