# Peer review of "Three-Month Outcomes in Cancer Patients with Superficial or Deep Vein Thrombosis in the Lower Limbs: Results from the RIETE Registry"

_cancers, 2023, doi:10.3390/cancers15072034_

Round 1

Reviewer 1 Report

The study is based of the RIETE registry which is one of the more well-known registries in this area of medicine, a key advantage of this study. I agree with the limitations raised in which the authors were unable to provide a more accurate entry of distance between SVT to deep venous system which may influence risk of propagation, as well as limited comments on how dose/type of anticoagulation therapy may have influenced outcomes. Some additional minor points:

1. How was DVT or SVT diagnosed? Would it be whole leg compression US or proximal? Are patients with SVT only, also radiographically cleared of DVT or is it presumed?

2. Paragraph 4 of discussion: "Unexpectedly, less than 5% of cancer patients and SVT and 20% of non-cancer patients with SVT were prescribed Fondaparinux in our cohort, many of them at lower than recommended doses. Fondaparinux is the only drug approved by the European Medicines Agency for the treatment of patients with isolated SVT." - rather than using "less than 5% etc", suggest use accurate figure - while this may be the case for EMA, other countries do use LMWH including enoxaparin (which was not included in the table) - I assume the doses listed in Table 3 are for dalteparin - pls clarify - as for the dose of fondaparinux, 7.5mg daily is the average "treatment" dose - using prophylactic dose, particularly in non-cancer patient without high risk is not unexpected. Pls comment on what is meant by recommended doses

3. Suggest rewording the conclusion - the term "half way" is unclear. Does the authors mean intermediate risk? Same comment applies to Fig 1

4. were any of the SVT characteristics eg length of SVT, treated or untreated predictive of propagation?

Author Response

  1. How was DVT or SVT diagnosed? Would it be whole leg compression US or proximal? Are patients with SVT only, also radiographically cleared of DVT or is it presumed?

ANSWER:

Many thanks for your positive comments.

As we stated in the Methods section (point 2.2. Inclusion criteria), “SVT was defined as the presence of a confirmed non-compressible venous segment by compression ultrasonography in a superficial vein without concomitant DVT or PE”.

Whether it was whole leg compression US or proximal only, it varied according to the criteria of the participating centers. RIETE is an international registry with over 200 participating centers in 26 countries, and it was not possible to standardize every detail across the centers.

  1. Paragraph 4 of discussion: "Unexpectedly, less than 5% of cancer patients and SVT and 20% of non-cancer patients with SVT were prescribed Fondaparinux in our cohort, many of them at lower than recommended doses. Fondaparinux is the only drug approved by the European Medicines Agency for the treatment of patients with isolated SVT." - rather than using "less than 5% etc", suggest use accurate figure - while this may be the case for EMA, other countries do use LMWH including enoxaparin (which was not included in the table) - I assume the doses listed in Table 3 are for dalteparin - pls clarify - as for the dose of fondaparinux, 7.5mg daily is the average "treatment" dose - using prophylactic dose, particularly in non-cancer patient without high risk is not unexpected. Pls comment on what is meant by recommended doses

ANSWER:

We modified the text according to your suggestion. It currently reads:

"Unexpectedly, only 4.5% of cancer patients and SVT and 20% of non-cancer patients with SVT were prescribed Fondaparinux in our cohort, most of them at lower than recommended doses for treatment (i.e., 7.5 mg daily). Fondaparinux is the only drug approved by the European Medicines Agency for the treatment of patients with isolated SVT."

As for your comment for the LMWHs in Table 3, we must clarify that all kind of LMWHs were used, including dalteparin, tinzaparin, enoxaparin or bemiparin. This is why we did not use the term “LMWH at lower than recommended doses” in the Table, as we used for Fondaparinux or the DOACs.

  1. Suggest rewording the conclusion - the term "half way" is unclear. Do the authors mean intermediate risk? Same comment applies to Fig 1

ANSWER:
Many thanks. We modified the text as requested. It currently reads:

“In conclusion, the natural history of SVT in cancer patients seems to be better than DVT in cancer patients but worse than SVT in non-cancer patients”

As for Figure 1, we reworded the text. It currently reads:

“The outcome of SVT in cancer patients was worse than SVT in non-cancer patients, but better than DVT in cancer patients”

4. were any of the SVT characteristics eg length of SVT, treated or untreated predictive of propagation?

ANSWER:

Unfortunately, this was not the aim of the study. But we very much appreciate your comment and will address this topic in a next forthcoming study.

Reviewer 2 Report

I have the following comments and questions about this manuscript:

1.) One thing I would like to know is if there is a difference between SVT in an axial vein (such as the GSV or SSV) and SVT isolated to varicosities.   As Table 1 suggests that the incidence of SVT in varicosities was between 45 and 60%, can the authors answer this question for me?  Should we think of SVT in varicosities the same as SVT in the GSV or SSV, and what about when there is a combination of SVT in both axial veins and varicosities?

 2.) On page 8, line 230, should the line read “Unexpectantly, less than 5% of cancer patients with SVT…..”, not “Unexpectedly, less than 5% of cancer patients and SVT……”.  The change is the word “with” for the word “and”.

3.) I do not understand the Conclusion on page 8, lines 255-260 and Figure 1.   What do the authors mean “SVT in cancer patients seems to be half way between DVT in cancer patients and SVT in non-cancer patients (Figure 1).   What does this refer to and what does this mean??

4.) I was surprised that patients with SVT were less likely to metastasis at baseline.  I would have expect the opposite.   Can the authors please address? 

Author Response

1.) One thing I would like to know is if there is a difference between SVT in an axial vein (such as the GSV or SSV) and SVT isolated to varicosities.   As Table 1 suggests that the incidence of SVT in varicosities was between 45 and 60%, can the authors answer this question for me?  Should we think of SVT in varicosities the same as SVT in the GSV or SSV, and what about when there is a combination of SVT in both axial veins and varicosities?

ANSWER:

Unfortunately, we do not gather this information in the RIETE database. But we agree with this reviewer and will include the exact site of the SVT as a new variable in the registry. This will help us to better learn about the natural history of patients with SVT, with- or without cancer.

 2.) On page 8, line 230, should the line read “Unexpectantly, less than 5% of cancer patients with SVT…..”, not “Unexpectedly, less than 5% of cancer patients and SVT……”.  The change is the word “with” for the word “and”.

ANSWER:

Many thanks. We modified the text, as requested.

3.) I do not understand the Conclusion on page 8, lines 255-260 and Figure 1.   What do the authors mean “SVT in cancer patients seems to be half way between DVT in cancer patients and SVT in non-cancer patients (Figure 1).   What does this refer to and what does this mean??

ANSWER:

This text has been modified according to the same comment by reviewer #1. It currently reads:

“In conclusion, the natural history of SVT in cancer patients seems to be better than DVT in cancer patients but worse than SVT in non-cancer patients”

4.) I was surprised that patients with SVT were less likely to metastasis at baseline.  I would have expected the opposite.   Can the authors please address? 

ANSWER:

We agree with this comment. There were a number of differences at baseline between cancer patients with SVT or DVT. Not only the presence of metastases, but also the elapsed time from cancer diagnosis, or the proportion of patients with breast, hematologic or bladder cancers in each side. This will be the topic for a next forthcoming study.